# The Impact of Treatment for Smoking on Breast Cancer Patients’ Survival

**DOI:** 10.3390/cancers14061464

**Published:** 2022-03-12

**Authors:** Akshara Singareeka Raghavendra, George Kypriotakis, Maher Karam-Hage, Seokhun Kim, Mazen Jizzini, Kareem S. Seoudy, Jason D. Robinson, Carlos H. Barcenas, Paul M. Cinciripini, Debu Tripathy, Nuhad K. Ibrahim

**Affiliations:** 1Department of Breast Medical Oncology, The University of Texas MD Anderson Cancer Center, Houston, TX 77030, USA; asraghavendra@mdanderson.org (A.S.R.); chbarcenas@mdanderson.org (C.H.B.); dtripathy@mdanderson.org (D.T.); 2Department of Behavioral Science, The University of Texas MD Anderson Cancer Center, Houston, TX 77030, USA; gkypriotakis@mdanderson.org (G.K.); maherkaram@mdanderson.org (M.K.-H.); seokhun.kim@uth.tmc.edu (S.K.); jdrobinson@mdanderson.org (J.D.R.); pcinciri@mdanderson.org (P.M.C.); 3Department of Psychiatry, The University of Texas MD Anderson Cancer Center, Houston, TX 77030, USA; 4Department of Medicine, State University of New York at Buffalo, Buffalo, NY 14260, USA; mazenjiz@buffalo.edu; 5Department of Psychiatry and Neurobehavioral Sciences, University of Virginia, Charlottesville, VA 22901, USA; kareem@virginia.edu

**Keywords:** smoking, breast cancer, overall survival, tobacco treatment program

## Abstract

**Simple Summary:**

This was a retrospective analysis of breast cancer patients who were self-identified as smokers at diagnosis and who were invited to participate in a comprehensive tobacco treatment program (TP) that provided pharmacotherapy and motivational counseling to quit smoking. Our study shows that quitting smoking is associated with improved survival among breast cancer patients who smoke across all tumor stages. In our survival analysis, tobacco abstainers were more likely than smokers to be alive with no evidence of disease (hazard ratio = 0.616 95%CI (0.402–0.945), *p* = 0.026). Comprehensive approach to address smoking cessation may prolong survival outcomes when started as early as the time of diagnosis.

**Abstract:**

Background: Smoking negatively affects overall survival after successful breast cancer (BC) treatment. We hypothesized that smoking cessation would improve survival outcomes of BC patients who were smokers at the time of diagnosis. Methods: This was a retrospective analysis of self-identified smokers with BC treated at The University of Texas MD Anderson Cancer Center. Patient demographics, date of diagnosis, tumor stage, tobacco treatment program (TP) participation, and time to death were extracted from our departmental databases and institutional electronic health records. We examined associations between tobacco abstinence status and survival using survival models, with and without interactions, adjusted for personal characteristics and biomarkers of disease. Results: Among all 31,069 BC patients treated at MD Anderson between 2006 and 2017, we identified 2126 smokers (6.8%). From those 2126 self-identified smokers, 665 participated in the TP, reporting a conservative estimate of 31% abstinence (intent-to-treat) 9 months into the program. Patients without reported follow-up abstinence status (including TP and non-TP participants) were handled in the analyses as smokers. Survival analysis controlled for multiple factors, including disease characteristics and participation in the TP, indicated that abstainers were more likely to be alive with no evidence of disease compared to non-abstainers (HR, 0.593; 95% CI, 0.386–0.911; *p* = 0.017). Conclusion: Our results suggest that quitting smoking is associated with improved survival among BC patients who were smokers at time of diagnosis across all tumor stages. Comprehensive approaches for smoking cessation in patients diagnosed with BC may prolong survival when started as early as the time of diagnosis.

## 1. Introduction

Smoking is associated with increased long-term all-cause lung- and breast-cancer-specific mortality, and negatively influences overall survival after breast cancer diagnosis [1,2]. As noted in the literature, non-smokers have better breast cancer survival compared to smokers [2]; however, there is a paucity of data on the impact of providing comprehensive tobacco treatment in changing this equation for those who could not quit on their own. Smokers who continue to smoke have a higher risk of lung cancer following breast cancer radiotherapy [3], while nonsmokers have minimal added risk of late radiation-induced lung cancer and cardiac mortality. Smoking is also associated with an increased breast cancer risk for women who began smoking before their first birth, suggesting a possible role for smoking in breast cancer tumorigenesis [4]. Additionally, multivariable-adjusted models with smoking status as a time-dependent variable have shown that breast cancer incidence was significantly higher among current and former smokers compared with never-smokers [5]. Smoking is also associated with a higher rate of breast cancer recurrence after partial mastectomy and radiotherapy [6]. Furthermore, continued smoking impairs wound healing, causes poor surgical outcomes, and increases the risk of postoperative complications among patients undergoing breast reconstruction surgery [7].

Providing assistance for breast cancer patients to quit smoking is a high-value intervention with the potential for lowering the risks of radiation-induced toxicity, lung cancer, and cardiac mortality [8] to approximately that of nonsmokers, as well as reducing the risk of local-regional breast cancer recurrence, breast reconstruction complications, and all-cause mortality [3]. Such assistance with a treatment for smoking is available through the MD Anderson comprehensive tobacco treatment program (TP). Established in 2006, the TP is funded via tobacco settlement money and provided at no cost to patients with the specific purpose of removing all barriers to care, in particular for breast cancer patients who lack the resources needed to pursue quitting in a fee-for-service program or with an addiction specialist. The TP is a personalized intervention offering motivational interviewing and cognitive behavioral counseling plus pharmacotherapy. Medications offered include nicotine replacement (patch or lozenge), bupropion, and varenicline, alone or in combination. Patients who show signs of psychiatric disorders are offered appropriate treatment [9,10,11]. Although patients can self-refer, the most common accrual channel is via provider referral followed by proactive outreach by program staff to anyone identified as a tobacco user in their medical record (an opt-out model). The TP has been described in more detail elsewhere [10,11].

Prior studies investigating the benefits of quitting smoking had a smaller sample size than our current study and focused on long-term all-cause mortality [2,6,12]. A pooled analysis by Pierce et al. [13] reported a poorer prognosis for breast cancer for lifetime smokers and for former heavy-smokers compared with never-smokers. We hypothesized that active participation in the TP and quitting smoking would prolong survival among breast cancer patients who were smokers at the time of their diagnosis.

## 2. Methods

### 2.1. Study Design and Patients

In a retrospective analysis using the prospectively maintained Breast Cancer Database Management System housed in the Department of Breast Medical Oncology at The University of Texas MD Anderson Cancer Center, we identified patients with breast cancer (*n* = 31,069) between 2006 (the year the TP was founded) and 2017 (the year corresponding to 5-year survivorship data available for analysis in the Breast Cancer Database). We identified those who were smokers at their initial visit to MD Anderson.

We reviewed the electronic medical records of these patients and extracted demographic characteristics, including ethnicity/race; menopausal status; body mass index; family history of breast and ovarian cancer in first- and second-degree relatives; tumor characteristics, including stage, biomarkers, and grade; treatment received (type of surgery, radiotherapy, and endocrine therapy).

Patients self-reported their race at the time of registration. Menopausal status was recorded at the diagnosis and was defined as pre-, peri-, or post-menopausal by the attending physician using the guidelines that were current at that time [14]. Women who were perimenopausal were grouped with women who were premenopausal. By the inclusion criteria, all patients, men or women, were at least 18 years old at diagnosis and received their initial cancer treatment and subsequent surveillance visits at MD Anderson between 2006 and 2017. The tumor stage was determined using the American Joint Committee on Cancer guidelines current at the date of diagnosis [15,16,17].

Biomarkers of tumors included positive or negative findings for estrogen receptor or progesterone receptor by immunohistochemistry using institutional cutoffs. Human epidermal receptor (HER2) status was assessed by immunohistochemistry or fluorescence in situ hybridization, when available, and deemed positive or negative on the basis of institutional cutoffs and guidelines that were current at the time of diagnosis [18].

From a total of 665 patients who entered the TP, those who were lost to follow-up at the TP’s 9-month outreach were considered smokers and were added to the non-abstinence group using a conservative intent-to-treat approach [19,20]. To conserve limited resources, the TP treats all patients who received their cancer treatment and subsequent surveillance visits at MD Anderson, but does not treat those who were present only for one visit (e.g., an initial consultation or a second opinion) [21]. Abstinence was defined as 7-day point prevalence, and self-reported data were systematically collected in real-time and saved in the TP departmental database using a timeline follow-back method [22], beginning with the baseline assessment at the end of treatment and then at 3-, 6-, 9-, and 12-month follow-up sessions. Carbon monoxide verification of abstinence was carried out at all in-person visits and had around 0.9 correlation with self-reporting [10].

Patients who were not TP participants (*n* = 1461) had their smoking status at 9 months after diagnosis imputed as non-abstinent, employing the intent-to-treat method common in the addiction literature [19,20].

The last follow-up date in the TP for patients in this study was 14 April 2020. This study was approved by MD Anderson’s Institutional Review Board, and a waiver for informed consent was obtained (PA18-0604). We conducted this analysis to evaluate the relationship between stopping smoking and breast cancer-specific survival (BCSS), which was the primary endpoint for the current study. We defined BCSS as the duration from the date of diagnosis of breast cancer until death due to breast cancer, termed dead with disease. Death was ascertained from death certificates, the institutional tumor registry, and hospital records for all patients with a cause of death. Similarly, we defined alive with no evidence of disease as complete remission with no clinical signs of cancer. Abstinent participants included smokers at baseline who reported abstaining from smoking at the 9-month follow-up session. The multivariate models were adjusted for age, race, menopausal status, cancer stage, tumor grade, receptor status (hormonal and HER2), histology, lymphatic and vascular invasion, type of surgery, receipt of neoadjuvant and adjuvant chemotherapy, receipt of radiotherapy, and TP participation. The median time to death, for those who died, was 1337 days with the 5% and 95% percentiles being 416 and 5956 days, respectively. The median time for all patients who either died or were lost to follow-up was 1599 days with the 5% and 95% percentiles being 413 and 5339 days, respectively.

### 2.2. Statistical Analysis

We used descriptive statistics to present associations of the patient characteristics with their abstinence status and survival status. For categorical variables, a χ^2^ test for abstinence status or a log-rank test of equality across strata for survival outcome was performed. For age, a univariate logistic regression for abstinence status, and a univariate proportional hazard (PH) regression model for survival, was used. We adopted the parametric Gompertz model for this analysis because it effectively assesses adult mortality rates [23,24]. The Gompertz distribution is basically a log-Weibull distribution with two parameters (shape and scale) and has been shown to provide a superior fit for survival after the diagnosis of breast cancer patients [25].

Since patients were not randomized into the two abstinence categories, there was a potential for bias due to confounders for survival status. To address this possibility, a propensity score (PS) for inclusion in the abstinent category was estimated using a logistic regression model including all demographic and disease-characteristic covariates that were identified in the descriptive statistics for their potential association with abstinence status. Using the PS, we calculated inverse probability weights (IPW) for each patient and applied them in the estimation of the final PH models. We did this to achieve approximate covariate balance between the two groups and eventually an approximately unbiased estimate of the abstinence effect [26,27]. The covariate balance for the PS was evaluated using standardized mean difference scores for each covariate after applying the IPW (i.e., adjusted scores); a conservative threshold of 0.1 was used to check whether the IPW achieved a satisfactory balance [28]. A doubly-robust multivariable Gompertz PH model with IPW was conducted with the main predictor and all the covariates to assess the effect of abstinence while correcting any residual bias due to the covariates [29,30,31]. This doubly-robust estimation accounts for confounding even in the case of mis-specification of the model (for example, see Funk et al. [29]).

For this model, stage 0 BC patients (*n* = 202) were excluded, since it is a less fatal disease, as were patients with the menopausal status of a male at diagnosis (*n* = 3) so as to prevent an unstable estimation due to a small total number with no death having been observed. Furthermore, given that our main predictor was abstinence status at 9 months, all patients with a time to death less than 9 months after diagnosis (*n* = 91) were excluded to prevent a bias in estimation. The PH assumption was tested and not violated using scaled Schoenfeld residuals from a Cox PH model with the same variables, both statistically and visually. In addition, there was no multicollinearity among the independent variables. Moreover, in order to assess the sensitivity and robustness of the results for the total sample related to the intent-to-treat imputation of non-abstinence status for the non-TP participants, we performed a complete case analysis with only TP participants for whom we had collected records of abstinence status.

## 3. Results

### 3.1. Univariate Analysis

Figure 1 presents the steps for the final cohort identification used for analysis (*n* = 2126) as a STROBE diagram. The abstinent group was composed of patients identified as abstinent from their participation in the TP, and the non-abstinent group included both patients in the TP who identified themselves as smokers at the program’s last follow-up and patients not in the TP program who were identified as smokers at baseline, and their smoking status was imputed as smoking. To calculate breast cancer-specific survival, we excluded patients that were alive with disease, dead without the disease, and dead due to any other cause.

Table 1 presents the comparison of individual characteristics with respect to abstinence status. Race was significantly associated with abstinence (*p* = 0.041). Pairwise comparisons revealed that the effect of race was exclusively attributed to differences in abstinence between white and black patients, with black patients experiencing higher abstinence rates (OR, 1.63; 95% CI, 1.17–2.26). Appendix A presents the comparison of individual level characteristics with respect to survival status (1769 were alive with no evidence of disease, and 357 were dead with disease).

The univariate analyses of log-rank tests and the PH model for the outcome determined that the following variables were associated with the risk of death due to disease, at the significance level of 0.05: (1) abstinence status, (2) stage of disease, (3) hormone receptor status, (4) HER2 status, (5) neoadjuvant chemotherapy, (6) adjuvant chemotherapy, (7) adjuvant hormonal therapy, (8) adjuvant radiotherapy, (9) tumor grade, (10) tumor histology, (11) lymphatic invasion, (12) vascular invasion, (13) surgery type, and (14) stage IV de-novo disease. On the other hand, the associations with the following variables were not significant: race, menopausal status, neoadjuvant hormonal therapy, laterality, and age at diagnosis. Since these non-significant covariates have been shown to be related to a breast cancer patient’s survival status in the literature [32,33], they were also included in the multivariable PH model below.

### 3.2. Survival

Before estimating the final models for BCSS, we estimated the PS for inclusion in the abstinent group for all participants using a logistic regression model to achieve covariate balance between the two abstinence groups for all covariates. After applying inverse probability weights (IPW) based on the PS, the adjusted standardized mean difference was lower than the threshold of 0.1 for all the covariates (see Appendix A), indicating that the IPW achieved a satisfactory covariate balance between the two groups.

Figure 2 illustrates the estimated survival functions for the two abstinence groups. The doubly-robust (doubly-robust because we included both the IPW and the covariates in the estimation) PH model revealed that abstinence decreased the risk of death by 42.8% compared with non-abstinence with all the covariates held constant (hazard ratio [HR], 0.572; z = −2.46; *p* = 0.014; 95% CI = 0.366–0.893; Table 2). Furthermore, as shown in Table 2, there was a 2.0% increase in the expected hazard of death when age was increased by one year. Other covariates associated with a higher risk were neoadjuvant chemotherapy, HER2-negative status (compared with HER-2 positive), and surgery of unknown category (compared with lumpectomy). In contrast, mastectomy (compared with lumpectomy), neoadjuvant hormonal therapy, adjuvant hormonal therapy, radiotherapy, and post-menopausal status were associated with a lower risk. No other significant associations were observed with any covariate, including cancer stage. The non-significant effect of the TP on survival in the models used in this paper should not be interpreted as evidence that the TP was ineffective in prolonging survival. On the contrary, the effect of the TP on survival is manifested through abstinence, as our findings show. Specifically, since all patients that abstained from smoking were TP participants, their increased survival (compared to non-abstinent) is evidence that the TP was effective in prolonging survival via abstinence. The observation that the effect of the TP in Table 2 was not significant is a result of regressor suppression, where a more immediate predictor can nullify the effect of a distant predictor when the distant and immediate predictors are highly correlated. To validate this assumption, we estimated the effect of the TP on survival after excluding abstinence as a predictor. The effect of participation in the TP on survival was significant (HR = 0.74; CI: 0.58, 0.96; *p* = 0.021), suggesting that participation in the TP could be effective for increasing the survival of breast cancer patients due to a higher abstinence rate [10].

To examine the sensitivity of the above significant results for the effect of abstinence on BCSS, we performed an additional analysis including only patients who participated in the TP. The results of the TP-only sample analysis confirmed the earlier results for the total sample analysis, providing support of the presence of an effect. The specific effect of abstinence status on BCSS for the TP-only sample was significant (HR, 0.582; z = −2.19; *p* = 0.029; 95% CI, 0.358–0.945; Table 3) using the same estimation method and covariate control as the analysis depicted in Table 2.

## 4. Discussion

In our retrospective analysis, breast cancer patients who abstained from smoking after their diagnosis had significantly improved survival outcomes. Smoking has been identified as an independent risk factor that affects breast cancer survival; however, several factors such as age, race, menopausal status, stage, grade, receptor status (hormone receptor, HER2), histology, lymphatic and vascular invasion, surgery, chemotherapy, and radiotherapy may all influence survival in patients with breast cancer. Our data analysis identified smoking cessation as consistently associated with improved survival outcomes after adjusting for all known possible confounding variables [10,11,34]. Hormone receptor status is an important prognostic factor in breast cancer; however, reports of whether the risk of developing hormone receptor-positive breast cancer depends upon smoking history remain inconsistent [35,36,37]. Since hormone receptor-positive disease may have a more indolent course, it may have skewed the influence of smoking cessation on overall survival. Thus, it is noteworthy that our analysis did not reveal a significant interaction between cancer predictive factors and abstinence and their effects on survival.

Smoking has been correlated with a lower efficacy for some cancer treatments and a greater frequency of late side effects of breast cancer therapy. Of particular importance is the lower risk of late radiation effects among patients who abstained from smoking [3]. Smoking also influences the immune tumor microenvironment, which in turn, affects the body’s response to chemotherapy [38]. A study among aromatase inhibitor-treated patients aged 50 years or older and were smokers at their preoperative visit found a greater risk of breast cancer events, distant metastasis, and death compared with nonsmoking patients [39]. Another study of breast cancer patients treated with partial mastectomy and radiotherapy showed a significantly higher recurrence rate among smokers than prior smokers or never-smokers; however, it is not known whether the recurrence was local or distant [6].

In addition to being a primary public health problem, smoking increases the financial burden on the health care system. A recent cost calculation for smoking cancer patients whose disease did not respond to first-line treatment estimated at least $10,000 in additional costs for smokers according to multiple models [40]. An increased financial burden can negatively affect treatment adherence and possibly even mortality [41].

Passarelli et al. addressed the impact of continued smoking compared with quitting on overall mortality and breast cancer-specific mortality, and found an increase in overall mortality for patients who continued to smoke with a numerical increase of 30% in breast cancer-specific mortality [12], but the increase was not statistically significant because of the small sample size. Nevertheless, those findings illustrated the potential benefit of smoking cessation for breast cancer patients.

Providing comprehensive tobacco treatment in the oncologic setting has resulted in sustained high abstinence rates for all patients, with or without cancer, as well as cancer survivors [10].

A retrospective study [2] found that all-cause mortality is related to continued smoking, and Jones et al. [42] found smoking was associated with an increased risk of breast cancer, but there was no intervention or treatment strategies in either of these studies as supported in our study. Other studies have reported on the challenges in helping breast cancer patients quit smoking with traditional approaches, and emphasized the importance of exploring new behavioral or pharmacological interventions [43], validating our comprehensive approach and its substantially improved abstinence rates. Furthermore, our intervention is integrated with a real-world oncology setting distinguished by real-time data collection and storage (not chart reviews), and a long-term follow-up for those treated to confirm abstinence up to one year later. A retrospective study of 124 breast cancer patients concluded that smoking is not assessed consistently in 30% of patients, and that only five smokers with breast cancer were referred specialized treatment, which highlights the importance of universal screening and the provision of comprehensive treatment, as in our study, resulting in higher abstinence and survivorship [44]. In particular, intervention makes a difference to the rate of quitting and improves survivorship above and beyond natural progression. In fact, our assumption that all patients who did not participate in the TP continued to smoke in our study was very conservative, and this might have biased the findings in the opposite direction (since those who quit on their own in the non-TP participants would have improved survivorship based on the above literature). Despite this assumption, we still found higher survivorship rates for our group of TP participants. Breast cancer patients who participated in the TP reported high rates of abstinence, as did patients with other tumor sites [10]. The TP is a well-established comprehensive program providing counseling, pharmacotherapy, treatment of co-occurring mental health disorders, and regular follow-ups for cancer patients seeking to quit tobacco use. As result, patients’ abstinence outcomes are much higher compared with minimal intervention strategies, brief advice, or referral to a quit line, supporting the value of a comprehensive approach to tobacco treatment for cancer patients [11]. Based on our data, we posit that oncology providers can improve their patients’ survivorship by being explicit with their recommendation to quit smoking and explaining the benefits of quitting, even at the time of cancer diagnosis. Ideally, a breast cancer treatment plan for a smoker would include referral to a comprehensive program; this would include careful patient screening for tobacco use, an individualized treatment plan, and adequate longitudinal support.

While our retrospective study design in a single large institution and our choice of data analyses may limit the generalizability of our conclusions, our findings contribute clinical granularity to the much needed body of literature on the impact of smoking and smoking cessation on breast cancer and survivorship after treatment. This study’s strength is in its naturalistic design and the provision of a tobacco treatment intervention within the cancer treatment setting that provides real-world evidence for its effectiveness. Although the adjusted multivariate models considered a large set of potential confounders, residual confounding is possible in a non-randomized study from other lifestyle factors (e.g., physical activity, alcohol or drug use, weight control) [45]. There could be other confounding factors like differences in the adherence to treatment or follow-ups for the smokers vs. the nonsmokers that will require prospective trials to address. Another limitation of our analysis is that there was no long-term follow-up on the smoking status of smokers who did not participate in the TP; however, our assumption that they continued smoking is based on our experience, the natural history of tobacco use disorder (addiction) [46], and a recent analysis of a large epidemiological survey from 2017 that reported around 12% of breast cancer patients continued to smoke after their diagnosis [47]. We classified patients who participated in the TP and were lost to follow-up as smokers; however, there are other possible reasons for not completing a follow-up session, among them: (1) having already quit and not needing any more help to remain abstinent, (2) moving or changing a phone number or address, and (3) death. The current study was performed in a real-life oncology setting that benefitted from the availability of highly annotated clinicopathologic data. Furthermore, the availability of detailed patient and initial treatment information collected in real-time during treatment at MD Anderson at both the Department of Breast Medical Oncology and the TP enabled us to demonstrate the favorable impact of smoking abstinence on survival among breast cancer patients.

## 5. Conclusions

This is a real-life oncology setting study that analyzed a highly annotated clinicopathologic data and allowed us to benefit from the availability of detailed patient and initial treatment information that was prospectively collected at MD Anderson at both the Department of Breast Medical Oncology and the TP of the Department of Behavioral Science. It demonstrated that tobacco cessation of patients with breast cancer diagnosis improved overall survival as a result of active intervention that resulted in smoking cessation.

## Figures and Tables

**Figure 1 cancers-14-01464-f001:**
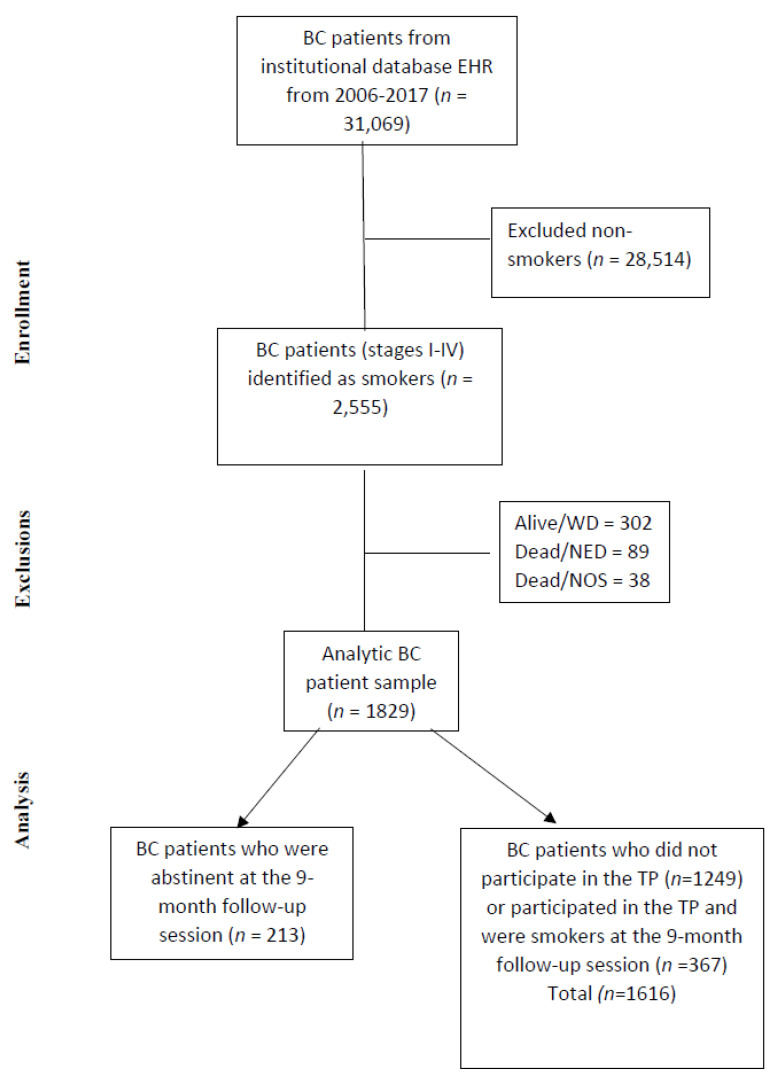
STROBE diagram. BC, breast cancer; NED, no evidence of disease; NOS, not otherwise specified; TP, tobacco treatment program; WD, with disease.

**Figure 2 cancers-14-01464-f002:**
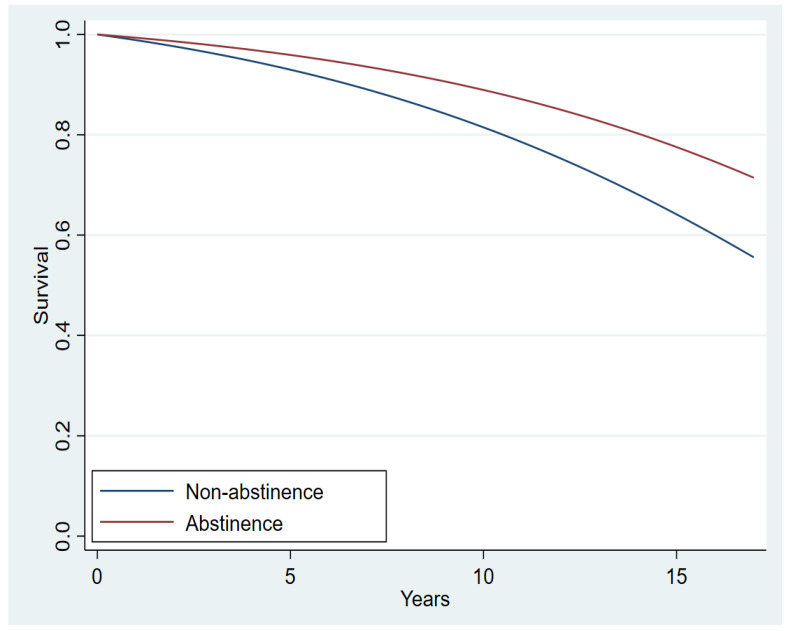
Estimated breast cancer-specific survival functions for the two abstinence groups (Hazard Ratio, 0.616; *p* = 0.026; 95% CI, 0.402–0.945).

**Table 1 cancers-14-01464-t001:** Demographic and disease characteristics of breast cancer patients with respect to tobacco abstinence status at 9-month follow-up.

Characteristic	Non-Abstinence No. (%) (*n* = 1616)	Abstinence No. (%) (*n* = 213)	*p*
Disease survival status			
Alive with no evidence of disease	1324 (81.93)	179 (84.04)	0.450
Dead with disease	292 (18.07)	34 (15.96)	
Age at diagnosis, y, mean (SD)	51.11 (10.88)	50.49 (10.16)	0.430
Race			0.038
White	1214 (75.12)	146 (68.54)	
Black	192 (11.88)	38 (17.84)	
Hispanic	168 (10.40)	20 (9.39)	
Asian	25 (1.55)	4 (1.88)	
Native American	4 (0.25)	0 (0.0)	
Other	13 (0.80)	5 (2.35)	
Stage			0.959
I	522 (32.30)	69(32.39)	
II	647 (40.04)	88 (41.31)	
III	359 (22.22)	46 (21.60)	
IV	88(5.45)	10 (4.69)	
Hormone receptor status			0.272
Positive	1253 (77.54)	158 (74.18)	
Negative	363(22.46)	55(25.82)	
HER2			0.001
Positive	273 (16.89)	24 (11.27)	
Negative	1314 (81.31)	178 (83.57)	
Not tested	29(1.79)	11(5.16)	
Menopausal status at diagnosis			0.368
Pre	676 (41.83)	96 (45.07)	
Post	940 (58.17)	117 (54.93)	
Neoadjuvant chemotherapy			0.409
Yes	568 (35.15)	81 (38.03)	
No	1048 (64.85)	132 (61.97)	
Neoadjuvant hormonal therapy			0.852
Yes	42 (2.60)	6 (2.82)	
No	1574 (97.40)	207 (97.18)	
Adjuvant chemotherapy			0.656
Yes	579 (35.83)	73 (34.27)	
No	1037 (64.17)	140 (65.73)	
Adjuvant hormonal therapy			0.879
Yes	1048 (64.85)	137 (64.32)	
No	568 (35.15)	76 (35.68)	
Adjuvant XRT			0.590
Yes	1001 (61.94)	136 (63.85)	
No	615(38.06)	77 (36.15)	
Grade			0.994
I	165 (10.21)	22 (10.33)	
II	747 (46.23)	99 (46.48)	
III	704 (43.56)	92 (43.19)	
Laterality			0.573
Right	771 (47.71)	106 (49.77)	
Left	845 (52.29)	107 (50.23)	
Histology			0.444
Invasive ductal	1369 (84.72)	174 (81.69)	
Invasive lobular	131 (8.11)	18 (8.45)	
Invasive mixed ductal/lobular	65 (4.02)	10 (4.69)	
Other	51 (3.16)	11 (5.16)	
Lymphatic invasion			0.292
Positive	329 (20.36)	50 (23.47)	
Negative	1287 (79.64)	163 (76.53)	
Vascular invasion			0.218
Positive	321 (19.86)	50 (23.47)	
Negative	1295 (80.14)	163 (76.53)	
Surgery type			0.349
Lumpectomy	668 (41.34)	97 (45.54)	
Mastectomy	874 (54.08)	108 (50.70)	
N/A (stage IV/not done)	55 (3.40)	4 (1.88)	
Unknown	19 (1.18)	4 (1.88)	
Stage IV de novo			0.622
Yes	89 (5.51)	10 (4.69)	
No	1527 (94.49)	203 (95.31)	

Abbreviations: XRT, radiotherapy.

**Table 2 cancers-14-01464-t002:** Multivariable proportional hazards model for breast cancer-specific survival among the intent-to-treat population (*n* = 1829).

Covariate	Hazard Ratio	*p*	95% CI, Lower	95% CI, Upper
Abstinence				
No	Ref			
Yes	0.572	0.014	0.366	0.893
TTP Participation				
	1.140	0.360	0.861	1.510
Race				
White	Ref			
Asian	0.917	0.781	0.499	1.686
Black	1.006	0.967	0.736	1.375
Hispanic	0.850	0.408	0.579	1.248
Native American	0.898	0.642	0.571	1.412
Others	1.024	0.968	0.320	3.275
Stage				
I	Ref			
II	2.827	0.000	1.736	4.603
III	8.115	0.000	4.645	14.176
IV	47.658	<0.001	16.891	134.463
Hormone receptor status				
Positive	Ref			
Negative	0.966	0.848	0.686	1.367
HER2				
Positive	Ref			
Negative	2.422	<0.001	1.669	3.515
Not tested	0.561	0.245	0.213	1.484
Menopausal status at diagnosis				
Pre	Ref			
Post	0.758	0.094	0.547	1.048
Neoadjuvant chemotherapy				
Yes	Ref			
No	0.737	0.097	0.514	1.056
Neoadjuvant hormonal therapy				
Yes	Ref			
No	2.144	0.032	1.067	4.302
Adjuvant chemotherapy				
Yes	Ref			
No	1.140	0.460	0.805	1.615
Adjuvant hormonal therapy				
Yes	Ref			
No	1.654	0.010	1.130	2.419
Adjuvant XRT				
Yes	Ref			
No	1.761	0.002	1.226	2.530
Grade				
I	Ref			
II	1.151	0.549	0. 726	1.825
III	1.487	0.108	0.916	2.415
Laterality				
Right	Ref			
Left	0.951	0.661	0.760	1.189
Histology				
Invasive ductal	Ref			
Invasive lobular	1.444	0.033	1.029	2.026
Invasive mixed	0.625	0.111	0.351	1.114
Other	0.432	0.027	0.205	0.909
Lymphatic invasion				
Positive				
Negative	0.460	0.038	0.221	0.958
Vascular invasion				
Positive				
Negative	1.128	0.750	0.538	2.365
Surgery type				
Lumpectomy	Ref			
Mastectomy	0.689	0.035	0.488	0.974
N/A (stage IV/not done)	1.281	0.361	0.753	2.180
Unknown	7.658	<0.001	4.346	13.494
Stage IV de novo				
Yes	Ref			
No	2.046	0.110	0.849	4.925
Age at diagnosis				
	1.016	0.039	1.001	1.031

Abbreviations: XRT, radiotherapy.

**Table 3 cancers-14-01464-t003:** Complete case analysis using the multivariable proportional hazards model for breast cancer-specific survival among patients who participated in the tobacco treatment program (*n* = 580).

Covariate	Hazard Ratio	*p*	95% CI, Lower	95% CI, Upper
Abstinence				
No	Ref			
Yes	0. 582	0. 029	0. 358	0. 945
Race				
White	Ref			
Asian	0.504	0.131	0.208	1.226
Black	1.215	0.459	0.726	2.032
Hispanic	0.718	0.465	0.296	1.742
Native American	N/A			
Other	10.840	<0.001	3.849	30.528
Stage				
I	Ref			
II	2.451	0.050	0.998	6.018
III	10.730	<0.001	3.873	29.729
IV	18.141	<0.001	4.856	67.764
Hormone receptor status				
Positive	Ref			
Negative	1.976	0.048	1.007	3.879
HER2				
Positive	Ref			
Negative	3.383	0.004	1.479	7.734
Not tested	0.828	0.753	0.258	2.662
Menopausal status at diagnosis				
Pre	Ref			
Post	0.639	0.129	0.359	1.139
Neoadjuvant chemotherapy				
Yes	Ref			
No	0.654	0.246	0.319	1.339
Neoadjuvant hormonal therapy				
Yes	Ref			
No	1.630	0.441	0.471	5.644
Adjuvant chemotherapy				
Yes	Ref			
No	1.262	0.465	0.676	2.355
Adjuvant hormonal therapy				
Yes	Ref			
No	0.845	0.655	0.406	1.760
Adjuvant XRT				
Yes	Ref			
No	1.973	0.060	0.971	4.011
Grade				
I	Ref			
II	0.918	0.786	0.496	1.699
III	0.777	0.500	0.373	1.619
Laterality				
Right	Ref			
Left	0.819	0.404	0.514	1.306
Histology				
Invasive ductal	Ref			
Invasive lobular	1.188	0.574	0.650	2.172
Invasive mixed	0.266	0.123	0.049	1.432
Other	0.723	0.565	0.240	2.177
Lymphatic invasion				
Positive				
Negative	0.568	0.267	0.210	1.539
Vascular invasion				
Positive				
Negative	0.758	0.583	0.283	2.034
Surgery type				
Lumpectomy	Ref			
Mastectomy	0.530	0.038	0.291	0.9656
N/A (stage IV/not done)	3.106	0.011	1.296	7.447
Unknown	7.574	0.006	1.812	31.658
Stage IV de novo				
Yes	Ref			
No	0.769	0.653	0.2449	2.417
Age at diagnosis				
	1.029	0.095	0.995	1.066

Abbreviations: XRT, is radiotherapy.

## Data Availability

The data that support the findings of this study are available from the corresponding author, upon reasonable request.

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
