# Peer review of "The Impact of Treatment for Smoking on Breast Cancer Patients’ Survival"

_cancers, 2022, doi:10.3390/cancers14061464_

Round 1

Reviewer 1 Report

An interesting piece of work has been presented for review linking breast cancer survival rates with smoking status. I had a few questions for the authors: 1) Why weren't survival rates assessed in non-smokers and those who had recently quit smoking? For comparison, it would be interesting to see the corresponding curve on the graph. 2) Lower survival rates in smokers may be associated with lower living standards, socioeconomic status, poorer attitudes towards one's health and, accordingly, with delayed compliance with medical recommendations and later seeking medical help. In the group of smokers, there are fewer people leading a healthy lifestyle. What do you think of it? 

Author Response

Reviewer 1:

An interesting piece of work has been presented for review linking breast cancer survival rates with smoking status. I had a few questions for the authors:

  • Why weren't survival rates assessed in non-smokers and those who had recently quit smoking? For comparison, it would be interesting to see the corresponding curve on the graph.

Response: This is indeed an important comparison that can reveal the magnitude of the difference in survival after diagnosis between smokers who quit and non-smokers. However, the population of interest for this study is breast cancer patients who were smokers at diagnosis. Specifically, this study examines how survival after diagnosis can be modified as a result of quitting in breast cancer patients who smoke. The main purpose and thrust of this paper is contributing to the literature about breast cancer and smoking but specifically looking into the impact of providing comprehensive tobacco treatment as intervention to help patients  quit smoking and as a result to improve breast cancer survival rates. For non-smokers the modifier (quitting) is not defined, so it would have been somewhat discordant and beyond the focus of this study to include non-smokers in the analysis. We are motivated, though, by this comment to collect data on non-smokers and compare their survival with smokers with breast cancer who quit, as a separate study.

Finally, guided by the reviewer’s comment, we included a  sentence in the introduction line 41 to 44 stating: “As noted in the literature, non -smokers have better breast cancer survival compared to smokers; however, there is paucity of data on the impact of providing comprehensive tobacco treatment in changing this equation for those who could not quit on their own” 1

2) Lower survival rates in smokers may be associated with lower living standards, socioeconomic status, poorer attitudes towards one's health and, accordingly, with delayed compliance with medical recommendations and later seeking medical help. In the group of smokers, there are fewer people leading a healthy lifestyle. What do you think of it?

Response: We appreciate this comment – and agree with the reviewer that lower survival rates in smokers are potentially  associated with lower living standards, and other contextual-level and individual-level characteristics by affecting treatment compliance and health.

In this study we used propensity score matching to adjust for several disease characteristics, such as stage and treatment, and individual-level characteristics, such as race.  Further, the fact that our tobacco cessation program was made free of charge to allow access to all patients regardless of their resources, therefore our patients are more likely to be with little or no resources to pursue quitting in a paid for program fee for service kind of setting or with an addiction specialist.

To a large extent, differences in disease severity and treatment compliance are reflecting differences in socio-economic status, and access to treatment and health behavior, as the reviewer points out. Thus, adjusting for differences in disease severity and treatment when comparing abstinent with non-abstinent, indirectly accounts for some of the background factors that the reviewer mentions.  We acknowledge that important information can be gained by also examining the direct relationship of SES, etc. on survival, not examined in this study.

We have included a sentence in the introduction line 59 to 64 stating to clarify this issue as “Such assistance with treatment for smoking is available through The MD Anderson Comprehensive Tobacco Treatment Program (TP), established in 2006, which is funded via tobacco settlement money and provided with no cost to patients; specifically with the purpose of removing all barriers to care  in particular for breast cancer patients who lack the resources needed to pursue quitting in a fee for service program or with an addiction specialist”.

Reviewer 2 Report

This topic has already been explored several times in the scientific community. This doesn't add any new information to the science. It's been long established that smoking cessation has positive outcomes in cancer. Just a few examples are below in BC.

https://academic.oup.com/jncics/article/1/1/pkx005/4209327

https://www.ncbi.nlm.nih.gov/pmc/articles/PMC8156674/

https://ascopost.com/News/58080?utm_source=TrendMD&utm_medium=cpc&utm_campaign=The_ASCO_Post_TrendMD_0&tid=PuMvYiByBMIXz7/HxaLxlwYV9Q/zxc6uCgLmYQJ/9WMfFIXF1517pRl4XKOwGTuLjUqpEQ==

https://breast-cancer-research.biomedcentral.com/articles/10.1186/s13058-017-0908-4

1) what's new in this study than already reported?

2) Significance of the study than already reported?

Author Response

Reviewer 2:

This topic has already been explored several times in the scientific community. This doesn't add any new information to the science. It's been long established that smoking cessation has positive outcomes in cancer. Just a few examples are below in BC.

Response:

We respectfully disagree with the reviewer and we believe that our study is unique in several respects. First, although a randomized controlled trial is not feasible in this context, we went beyond and above any other study has done in the past to balance the two groups (abstinent vs. non-abstinent) using a plethora of disease characteristics and individual variables. We used both propensity score matching and covariate adjustment to account for residual imbalance after matching. Previous studies have not used methods to balance the comparison groups beyond covariate adjustment.  Second, we tested several interactions of our main predictor (abstinence status) and important covariates, for example stage. That is, formally tested heterogeneity of the effect of abstinence on survival by a series of interaction terms. No other study has examined the possibility that the effect reported may vary by subgroup. Third, although the documented abstinence is through self-report, as it is in all other studies, in our study we used a random sample for self-report validation via biochemical CO verification. The biochemical verification showed high agreement (> 90%) between self-report and CO level. Finally, we mention several aspects that our unique in our study as we reflect on the references the reviewer kindly listed.

We thank the reviewer for their suggestions and providing us with an opportunity to comment on the below papers

https://academic.oup.com/jncics/article/1/1/pkx005/4209327

This editorial itself acknowledges the gaps  in the literature that our data and study design fills, as it is a real-world follow-up with data collected and saved in real-time on those treated (not by a later chart review). Moreover, our data show that formal comprehensive smoking cessation treatment do produce substantial abstinence which in turn correlates with improved survivorship, while prior studies as cited in the above editorial:

“ Postdiagnosis assessments of smoking habits are frequently missed, both in the clinic (3) and as part of follow-up in longitudinal research studies (4), making it a challenge to estimate any association with change in smoking behavior.”

“Unfortunately, cessation support for breast cancer survivors who are unwilling or unable to stop smoking has proved challenging. A clinical trial of physician-based smoking cessation guidelines, which included some women with advanced-stage breast cancer, did not result in a statistically significant change in quit rates six or 12 months postintervention (19). Specific to breast cancer, a clinical trial in Denmark of a one-time personalized cessation support and nicotine replacement therapy within a week prior to breast cancer surgery did not reduce postoperative complications relative to standard care (20). These studies reinforce the notion that new behavioral or pharmacological approaches to smoking cessation will require intense, long-term, continual patient support.”

We have included this paper in our discussion line 302 and mentioned as follows “Other studies have reported on the challenges  in helping breast cancer patients’ quit smoking with traditional approaches, and emphasize the importance of exploring new behavioral or pharmacological intervention;  giving validity to our comprehensive approach  resulting in substantially improved abstinence rates. Further, our intervention is integrated in as in real world oncology setting distinguished by real-time data collection and storage (not a chart review) and long-term follow-up on those treated to confirm abstinence up to one year after2

https://www.ncbi.nlm.nih.gov/pmc/articles/PMC8156674/

The authors in this paper conclude that smoking is not assessed consistently in 30% of their patients and only 5 smokers with breast cancer were sent to treatment, here again our study emphasizes the importance of universal screening and offering comprehensive treatment to everyone and show that it does result in higher abstinence rates.

The authors of the above paper reported: “A total of 1234 patients were included in the study. Smoking status at diagnosis was missing from electronic health records in 32% of cases, including 13% of patients who smoke. Only 20% of the 197 patients currently smoking at diagnosis recalled having a discussion about smoking with a healthcare professional. Radiotherapists and surgeons were more likely to talk about complications than other practitioners. The main type of information provided was general advice to stop smoking (n = 110), followed by treatment complications (n = 48), while only five patients were referred to tobaccologists. Since diagnosis, 33% (n = 65) of the patients currently smoking had quit. Patients who quit had a lower alcohol consumption, but no other factor was associated with smoking cessation. The main motivation for tobacco withdrawal was the fear of BC relapse (63%). This study highlights room for improvement in the assessment of smoking behavior.

We have included this paper in our discussion line 308  and mentioned as follows “ A retrospective study of 124 breast cancer patients concluded that smoking is not assessed consistently in 30% of patients and only 5 smokers with breast cancer were referred  to specialized treatment,  which highlights the importance of universal screening and the provision of comprehensive treatment as  in our study resulting in higher abstinence and survivorship”.3

https://ascopost.com/News/58080?utm_source=TrendMD&utm_medium=cpc&utm_campaign=The_ASCO_Post_TrendMD_0&tid=PuMvYiByBMIXz7/HxaLxlwYV9Q/zxc6uCgLmYQJ/9WMfFIXF1517pRl4XKOwGTuLjUqpEQ==

This reference is an ASCO post describing the Parada et. Al. paper, which is an important retrospective analysis of a large cohort showing that all-cause mortality is related to continuing smoking compared with non-smokers and those who quit after being diagnosed; however it is a natural progression of the disease with no intervention, only those who quit on their own were compared 5 years later with never smokers and continued smokers, but did not find significance for breast cancer mortality per se as the confidence interval crossed below 1 (specifically 0.79-3.23). While our study had a comprehensive intervention that led to specifically improved survivorship form breast Cancer mortality.

Study Results

The researchers found that compared with never smokers, the risk of all-cause mortality was elevated among the 19% of at-diagnosis smokers (HR = 1.69, 95% CI = 1.36–2.11), those who smoked 20 or more cigarettes per day (HR = 1.85, 95% CI = 1.42–2.40), women who had smoked for 30 or more years (HR = 1.62, 95% CI = 1.28–2.05), and women who had smoked 30 or more pack-years (HR = 1.82, 95% CI = 1.39–2.37). Risk of all-cause mortality was further increased among the 8% of women who were at-/postdiagnosis smokers (HR = 2.30, 95% CI = 1.56–3.39) but was attenuated among the 11% of women who quit smoking after diagnosis (HR = 1.83, 95% CI = 1.32–2.52). Compared with never smokers, breast cancer–specific mortality risk was elevated 60% (HR = 1.60, 95% CI = 0.79–3.23) among at-/postdiagnosis current smokers, but the confidence interval included the null value and elevated 175% (HR = 2.75, 95% CI = 1.26–5.99) when the researchers considered postdiagnosis cumulative pack-years.

https://breast-cancer-research.biomedcentral.com/articles/10.1186/s13058-017-0908-4

This study here does not describe intervention, while it is based on a large sample size in UK (over 100K)

We have included this paper in our discussion line 299 and mentioned as follows “A retrospective study4 found that all cause mortality is related to continuing smoking and Jones et al. found smoking associated with increased risk of breast cancer, but there was no intervention or treatment strategies in both these studies as supported in our study.”

1)           what's new in this study than already reported?

The unique qualities of our study is the propensity score matching which none of the previous studies have done to balance the groups to simulate randomization in a clinical trial with a plethora of cancer-specific characteristics, evaluation of interactions of stage with abstinence on survival and other interactions with a reliable (real-time collection and storage of data) and conservative measurement of abstinence, allowed us to adjust the survival hazard ratios associated with  abstinence. 

In particular, a comprehensive treatment intervention made a difference in rates of quitting and improved survivorship above and beyond natural progression, in fact assuming that all those who did not participate in tobacco treatment program continued to smoke is very conservative and it might bias the findings in the opposed direction (as those who quit on their own in the non-treated sample would have improved survivorship based on the above literature) as some of them may have quit and despite our assumption that they all were smokers we still found higher survivorship rates in our group of confirmed quitting.

We have included a sentence in the discussion line 312 as “In particular, intervention makes a difference in rates of quitting and improves survivorship above and beyond natural progression, in fact assuming that all those who did not participate in tobacco treatment program continued to smoke is very conservative and it might bias the findings in the opposed direction (as those who quit on their own in the non-treated sample would have improved survivorship based on the above literature) as some of them may have quit and despite our assumption that they all were smokers we still found higher survivorship rates in our group of confirmed quitting.”

2) Significance of the study than already reported.

We thank the reviewer for their comment and would like to mention the naturalistic nature of the study and that it was achieved within the cancer treatment setting real world evidence and effectiveness.  

We have included a sentence in the discussion line 335 as “This study’s strength is in its naturalistic nature and in providing the tobacco treatment intervention within the cancer treatment setting, which is real world evidence and effectiveness”.

  1. Parada, H.; Bradshaw, P.T.; Steck, S.E.; Engel, L.S.; Conway, K.; Teitelbaum, S.L.; Neugut, A.I.; Santella, R.M.; Gammon, M.D. Postdiagnosis changes in cigarette smoking and survival following breast cancer. JNCI cancer spectrum 2017, 1.
  2. Passarelli, M. N., & Newcomb, P. A. (2017). Survival benefits of smoking cessation after breast cancer diagnosis. JNCI Cancer Spectrum1(1), pkx005.
  3. Nicolas, M., Grandal, B., Dubost, E., Kassara, A., Guerin, J., Toussaint, A., ... & Hamy, A. S. (2021). Breast Cancer (BC) Is a Window of Opportunity for Smoking Cessation: Results of a Retrospective Analysis of 1234 BC Survivors in Follow-Up Consultation. Cancers13(10), 2423.
  4. Parada Jr, H., Bradshaw, P. T., Steck, S. E., Engel, L. S., Conway, K., Teitelbaum, S. L., ... & Gammon, M. D. (2017). Postdiagnosis changes in cigarette smoking and survival following breast cancer. JNCI cancer spectrum1(1), pkx001.
  5. Jones, M. E., Schoemaker, M. J., Wright, L. B., Ashworth, A., & Swerdlow, A. J. (2017). Smoking and risk of breast cancer in the Generations Study cohort. Breast Cancer Research19(1), 1-14.

Round 2

Reviewer 2 Report

Thanks for explaining all the concerns.